# Burnout among medical students of a medical college in Kathmandu; A cross-sectional study

**Dhan Bahadur Shrestha**[1]*, **Nagendra Katuwal**[2], **Ayush Tamang**[3], **Agrima Paudel**[3], **Anu Gautam**[3], **Muna Sharma**[3], **Ujwal Bhusal**[3], **Pravash Budhathoki**[4]

**1** Department of Emergency Medicine, Mangalbare Hospital, Urlabari, Nepal, **2** Department of Psychiatry, Shree Birendra Hospital, Kathmandu, Nepal, **3** Nepalese Army Institute of Health Sciences, College of Medicine, Kathmandu, Nepal, **4** Department of Emergency Medicine, Dr. Iwamura Memorial Hospital, Bhaktapur, Nepal

* medhan75@gmail.com

## Abstract

### Background

Medical students are more prone to burnout than the general population and students of other faculties due to the demanding nature of medical education with limited time and resources. Burnout has a negative impact on the academics and personal life of the students which can continue into their professional life and ultimately hamper patient care. The purpose of this study is to determine the prevalence of burnout among medical students of a medical college and find its association with age, gender, and year of study.

### Materials and methods

This cross-sectional study was conducted among medical students of a medical college in Kathmandu, Nepal from 14 January to 7 March, 2021. Stratified sampling followed by a simple random sampling technique was employed to select study participants. Data was collected through a self-administered questionnaire using the English version of the Oldenburg Burnout Inventory adapted for students (OLBI-S) and analyzed in STATA version 15.

### Results

The prevalence of burnout was found out to be 65.9% (n = 229). And of the remaining, 12.7% were exhausted, 11.4% were disengaged and 10.0% were neither exhausted nor disengaged. Burnout had no significant association with age in years, gender, and year of study.

### Conclusions

This study shows an alarming prevalence of burnout in almost two-thirds of medical students. These results indicate the necessity of employing effective strategies by relevant authorities for the mental well-being of future physicians. Further multicenter prospective studies are required for a better understanding of the prevalence and associated factors of burnout.

**Data Availability Statement:** All relevant data are within the paper and its Supporting information files.

**Funding:** The author(s) received no specific funding for this work.

**Competing interests:** The authors have declared that no competing interests exist.

## Introduction

Initially, burnout was defined for employees who work with other people, as a syndrome characterized by exhaustion, depersonalization, and reduced professional efficacy [1]. However, over the years, empirical research has shown that burnout is caused due to high job demands and/or lack of job resources [2] regardless of the job that they do [3, 4]. Similar to employees, students are also subject to burnout as they have to fulfill certain tasks such as attending classes, completing assignments within deadlines, and passing exams [5, 6].

In 2007, Demerouti et al. proposed Oldenburg Burnout Inventory (OLBI) considering exhaustion and disengagement to be the two core dimensions of job burnout [7, 8]. Reis et al. then adapted OLBI to measure academic burnout (OLBI-S) and defined it as a phenomenon that is characterized by feelings of (emotional, physical, and cognitive) exhaustion due to the demands of studying and an attitude of withdrawal and detachment from one's studies [9].

Numerous studies have shown that medical students experience high levels of burnout due to a highly stressful environment, competitiveness, excessive workload, sleep deprivation, peer pressure, and many other personal, curricular, institutional, and affective factors [10]. Studies on medical students have indicated the development of stress and burnout in preclinical medical education and its progression into clinical years [11, 12]. Burnout in medical students has been associated with depression, sleep deprivation, thoughts of dropping out, suicidal ideation, and substance abuse [10, 13–15]. It can also mitigate cognitive capabilities such as memory, interpretation of information, and skill acquisition [16].

Furthermore, the burnout of medical students can also lead to burnout after they become a doctor [17] which may result in unprofessionalism, poor quality of patient care, medical errors, suicidal ideation, and attrition, and be a factor in substance abuse and relationship difficulties [15, 18]. Many studies have found that the prevalence of burnout among medical students, residents, and physicians is as much as 50% in the US [19, 20]. Around the globe prevalence rates for medical students burnout ranges from 7.0% to 75.2%, depending on country-specific factors, applied instruments, cutoff criteria for burnout symptomatology [21]. However, burnout has not been adequately explored among the medical students of Nepal. Therefore, this cross-sectional study aims to find out the prevalence of burnout among medical students of a medical college in Nepal.

## Materials & methods

### Study design and settings

A cross-sectional study was done on undergraduate medical students of the Nepalese Army Institute of Health Sciences (NAIHS)-College of Medicine from 14 January to 7 March 2021. Since physical classes after COVID-19 lockdown resumed serially at a different time for different year students, it took 3 months for data collection. The list of students studying each year and their respective email addresses were obtained from the academic record section, and this was used as a sampling frame. Each of the randomly selected students was sent an email containing a link to Google Forms with the objectives explained and the written consent form was attached to the form itself. They were also provided with a unique random number which was to be filled in the Google forms to ensure the anonymity of the participants. Single response from each student was ensured via Google Forms setting by choosing 'Limit to 1 response'. The participants who did not respond to the email were sent multiple follow-up emails requesting to fill the Google form until the last day of acceptance of response i.e., 7 March, 2021.

## Study sample

The total number of students was 565 and excluding 5 research members from the 3rd year, the total population size was 560. Using Cochran's formula for sample size collection from Sampling Techniques [22].

$$
\begin{aligned}
n &= z^2 \times (p \times q) \div e^2 \\
&= 1.91.96^2 \times (0.49 \times 0.51) \div 0.05^2 \\
&= 384
\end{aligned}
$$

Where,
 n = Sample size
 z = 1.96 at 95% CI
 p = 0.49 (taken 0.49 based on 48.8% prevalence of burnout in a similar Nepalese study)
[23].
 q = 1-p
 e = marginal error, 5%

$$
\begin{aligned}
\text{Adjusted sample size} &= n \div \{1 + (n - 1) \div \text{N}\} \\
&= 228
\end{aligned}
$$

Where, N = Target population = 560

Therefore, the calculated sample size was 228, and then considering the 5% of non-response rate the sample size obtained was 239.

After calculation of sample size, a stratified sampling technique was employed using students' class year as a stratum. First, the total sample size (i.e., 239) was proportionally allocated for each stratum (First-year to Fifth-year), then a simple random sampling method using random number generator software was applied to obtain the study participants from each class year. This was done to address the potential selection bias and response bias (detail of procedure available as S1 File).

## Study instrument

Maslach Burnout Inventory (MBI) is the most commonly used scale for measuring burnout. Its three factors are depersonalization, emotional exhaustion, and personal accomplishment. However, studies have shown MBI's third factor to perform weakly and have recommended using a two-factor model [24]. Aptly, OLBI only focuses on the two dimensions (disengagement and exhaustion) and has both positively and negatively worded questions. Though OLBI is not widely used in the context of developing countries like ours, OLBI is more accessible as it is free to use unlike MBI [8]. Copenhagen Burnout Inventory (CBI)'s third scale, client/patient-related burnout, was not appropriate for preclinical medical students (1st and 2nd years) [25].

Hence, a self-administered questionnaire containing demographic variables and OLBI-S was used for the study (available as S2 File).

The demographics contained 3 items (age, gender, and academic year). OLBI-S measures burnout in two dimensions: exhaustion and depersonalization. Both subscales consist of eight items each, out of which four are positively worded and four negatively worded [9]. We recorded the participants' responses to the items on a 4 point Likert scale ranging from 1 (strongly agree) to 4 (strongly disagree).

## Analytical strategy

Only the completely filled questionnaires were included in our analysis. Both subscales of OLBI-S, exhaustion, and depersonalization, consist of four negatively worded items that have to be reversed i.e., from 1 to 4, 2 to 3, 3 to 2, and 4 to 1 so that higher scores correspond to higher burnout.

The mean of each subscale was taken and the cutoff scores of $\geq 2.25$ were used for exhaustion and scores of $\geq 2.1$ for disengagement to classify the participants into the following groups [26]:

- Low exhaustion and low disengagement (non-burnout group)

- Low exhaustion and high disengagement (disengaged group)

- High exhaustion and low disengagement (exhausted group)

- High exhaustion and high disengagement (burnout group)

Analysis was performed using STATA version 15. Simple frequency distribution was presented appropriately in the table and text. Mean score presented for each scale variable and subscale where appropriate. Binary logistic regression was carried with full-blown burnout as an outcome of interest (with 1 = yes and 0 = no). Age, year of study, and gender were independent variables and burnout was a dependent variable. Internal consistency of scale for this study was assessed using Cronbach's Alpha. The value of Cronbach's Alpha for 16 items of scale tested was 0.777 indicating a good level of internal consistency.

## Ethical consideration

Ethical clearance was obtained from the Nepalese Army Institute of Health Sciences Institutional Review Committee (NAIHS-IRC) before collecting data from participants (**Ref no: 365**). All the respondents were informed about the aims and objectives of the study by including the written consent form in the questionnaire itself. The participants were aware that their participation was voluntary. The confidentiality of the participants was ensured.

## Results

We mailed 239 students, among which 229 students completed the questionnaires. The mean age of the participants was 22.05 ± 1.84 years (median 22, IQR 21–23). Out of 229 respondents, 148 (64.6%) were males and 81 (35.4%) were females. Similarly, 53 (23.1%) were from the 5th year, 47 (20.5%) from the 4th year, 45 (19.7%) from the 3rd year, 40 (17.5%) from the 2nd year, and 44 (19.2%) from 1st year. Table 1 summarizes the response to the individual questions of OLBI-S.

The mean of the total score under the disengagement subscale was 2.287±0.349 (Median: 2.375; IQR: 2.125–2.500) and the mean of the total score under the exhaustion subscale was 2.504±0.428 (Median: 2.500; IQR: 2.250–2.813). Likewise, the mean total score under the disengagement subscale was 18.297±2.7955 (Median: 19; IQR: 17–20) and the mean scoring under the exhaustion subscale was 20.031±3.4264 (Median: 20; IQR: 18–22.5).

Following the previously mentioned cutoff values, 151 (65.9%) respondents fell under the burnout group, 26 (11.4%) fell under the disengaged group, 29 (12.7%) fell under the exhausted group whereas 23 (10.0%) fell under the non-burnout group. Fig 1 summarizes the burnout categories of the participants. The prevalence of burnout in our study was found to be 70.5% in the 1st year, 62.5% in the 2nd year, 71.1% in the 3rd year, 72.3% in the 4th, and 54.7% in the 5th year.

**Table 1. Response to individual questions.**

| Positively worded questions | Strongly agree n (%) | Agree n (%) | Disagree n (%) | Strongly disagree n (%) | Sum [Mean ±SD] |
|---|---|---|---|---|---|
| I always find new and interesting aspects in my studies. | 47(20.5%) | 159 (69.4%) | 16(7%) | 7(3.1%) | 1.926±0.6274 |
| I find my studies to be a positive challenge. | 68(29.7%) | 148 (64.6%) | 11(4.8%) | 2(0.9%) | 1.769±0.572 |
| I feel more and more engaged in my studies. | 24(10.5%) | 131 (57.2%) | 69(30.1%) | 5(2.2%) | 2.240±0.6616 |
| This is the only field of study that I can imagine myself doing. | 36(15.7%) | 90(39.3%) | 87(38.0%) | 16(7.0%) | 2.362±0.8296 |
| I can usually manage my study-related workload well. | 21(9.2%) | 123 (53.7%) | 72(31.4%) | 13(5.7%) | 2.336±0.7228 |
| I can tolerate the pressure of my studies very well. | 24(10.5%) | 141 (61.6%) | 57(24.9%) | 7(3.1%) | 2.205±0.6601 |
| After a class or after studying, I have enough energy for my leisure activities. | 19(8.3%) | 101 (44.1%) | 93(40.6%) | 16(7.0%) | 2.463±0.7461 |
| When I study, I usually feel energized. | 24(10.5%) | 138 (60.3%) | 62(27.1%) | 5(2.2%) | 2.210±0.648 |
| **Negatively worded questions** | **Strongly agree n (%)** | **Agree n (%)** | **Disagree n (%)** | **Strongly disagree n (%)** | **Sum [Mean ±SD]** |
| Over time, one can become disconnected from this type of study. | 26(11.4%) | 141 (61.6%) | 55(24.0%) | 7(3.1%) | 2.812±0.6653 |
| Lately, I tend to think less about my academic tasks and do them almost mechanically. | 10(4.4%) | 117 (51.1%) | 83(36.2%) | 19(8.3%) | 2.515±0.7108 |
| It happens more and more often than I talk about my studies in a negative way. | 7(3.1%) | 66(28.8%) | 128(55.9%) | 28(12.2%) | 2.227±0.6952 |
| Sometimes I feel sickened by my studies. | 11(4.8%) | 108 (47.2%) | 82(35.8%) | 28(12.2%) | 2.445±0.768 |
| There are days when I feel tired before I arrive in class or start studying. | 46(20.1%) | 136 (59.4%) | 38(16.6%) | 9(3.9%) | 2.956±0.7242 |
| After a class or after studying, I tend to need more time than in the past to relax and feel better. | 33(14.4%) | 98(42.8%) | 88(38.4%) | 10(4.4%) | 2.672±0.7736 |
| While studying, I often feel emotionally drained. | 21(9.2%) | 102 (44.5%) | 89(38.9%) | 17(7.4%) | 2.555±0.7625 |
| After a class or after studying, I usually feel worn out and weary. | 21(9.2%) | 118 (51.5%) | 75(32.8%) | 15(6.6%) | 2.633±0.7410 |

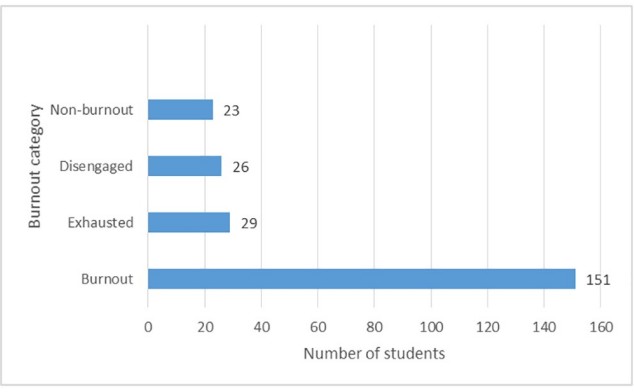

**Fig 1. Burnout categories among participated medical students.**

**Table 2. Binary logistic regression for the occurrence of burnout.**

| Burnout outcome | Adjusted OR | Std. Err. | z | P>|z| | [95% Conf. Interval] | |
|---|---|---|---|---|---|---|
| Age (years) | 1.176 | 0.169 | 1.13 | 0.260 | 0.887 | 1.560 |
| Gender | | | | | | |
| Female[®] | | | | | | |
| Male | 0.984 | 0.302 | -0.05 | 0.957 | 0.538 | 1.797 |
| Year of study | | | | | | |
| 3rd year[®] | | | | | | |
| 1st Year | 1.310 | 0.710 | 0.50 | 0.618 | 0.453 | 3.790 |
| 2nd Year | 0.804 | 0.391 | -0.45 | 0.653 | 0.310 | 2.087 |
| 4th Year | 0.880 | 0.432 | -0.26 | 0.794 | 0.336 | 2.304 |
| 5th Year | 0.355 | 0.185 | -1.99 | 0.047 | 0.128 | 0.985 |
| Constant | 0.072 | 0.223 | -0.85 | 0.397 | 0.0001 | 31.856 |

Binary logistics regression for outcome burnout did not differ across age in years, gender, and year of study. Table 2 summarizes the Binary logistic regressions for the occurrence of burnout.

## Discussion

Our study aimed to find the prevalence of burnout among undergraduate medical students and its association with demographic factors such as age, sex, and year of study. The prevalence of burnout among undergraduate medical students in our study was 65.9%, which is similar to the previous studies [12, 27, 28]. In several studies conducted in the US, the prevalence of burnout was found to be as much as 50% [19, 20]. However, according to Frajerman et al., the worldwide burnout prevalence for medical students lies around 44% [29]. And, the burnout calculated using Oldenburg Burnout Inventory (OLBI), ranged between 47.0 and 53.0% [21].

There is a considerable degree of heterogeneity in the prevalence of burnout among medical students. This is due to the use of different scales for measuring burnout and also differences in criteria for categorizing burnout within the same scale. In addition to this, the divergence in the prevalence may also be due to a multitude of factors like differences in curriculum, teaching methods, available facilities, and other institutional and personal factors.

Our study found the prevalence of burnout to be considerably higher than a previous study conducted in Nepal which reported the prevalence to be 48.8% [23]. This disparity in results could be due to the different instruments (CBI vs. OLBI) used to measure burnout. The higher prevalence in our study may also be due to the timing of the study period, which was a month after classes resumed post-COVID-19 lockdown. Still, multiple factors of the ongoing pandemic may have varying effects on the prevalence of burnout for different individuals. Furthermore, the higher prevalence of burnout may be due to the forthcoming final board examinations.

Regarding demographic data, there is no significant association of gender with a burnout in our study which is similar to the findings of some prior studies [30–32] but contrary to the studies which show high burnout in females compared to males [12, 33]. Also, no significant association was found between different age groups of students, which is similar to the findings of a previous study done by Dyrbye (2014) [18] and contrary to the findings of the previous study by O'Connor (2018) [34].

Some prior studies have also shown an increasing trend in the prevalence of burnout throughout their medical studies [35–37]. But we didn't find any correlation between burnout

and year of study. This may signify the hectic schedule, competitiveness, highly stressful environment, excessive workload, sleep deprivation, peer pressure, and many other personal, curricular, institutional, and affective factors throughout the years of medical education.

Among the preclinical year students (1st and 2nd years), the prevalence of burnout was found to be higher in the first year than in the second. The first-year medical students have a high level of stress even in the stage before the admission to medical school because many of them need gap year(s) to prepare for the medical entrance exams and gain acceptance to medical school. After admission, coping up with an entirely new environment away from home, living on their own, competitiveness, lack of leisure time or activities can be various contributory factors for high burnout in first-year medical students. Some students might have joined medical school under parental or peer pressure, or with the desire for higher socioeconomic status. They may become demotivated after knowing the reality of medicine and develop more negative attitudes towards medicine as a career [23, 38, 39]. Hence, they may be more likely to develop burnout in the long term.

Among the clinical year students (3rd, 4th, and 5th years), the prevalence of burnout among fifth-year students was relatively lower, yet insignificant, than the third and fourth-year students. This is contrary to the expected result of a higher prevalence of burnout among final year students due to their heavier curriculum and examination load. The higher prevalence of third and fourth-year students could be due to their transition from basic sciences into clinical sciences. Preclinical years consist mostly of didactic lectures in classrooms with little exposure to patients and clinical settings. Despite more rigorous academics and examinations, fifth-year students have to appear for examinations of subjects which they have been studying since their earlier clinical years (third and fourth-year). Additionally, they have been well acquainted with the clinical studies and examination pattern and could be less anxious about it.

However, only a longitudinal study of the same batch of students throughout their 1st to 5th year of study would give a clearer picture of the trend of burnout in different academic years. At an individual level, students should be encouraged to incorporate various self-care techniques into their daily lives such as a nutritious diet, regular exercise, restful sleep, making a balance between study and leisure, the practice of self-compassion, and being aware of their emotional needs [40]. Similarly, positive coping and hobbies help to overcome the stressors of medical studies. Some common practices can be mindfulness, yoga, listening to music, reading books, and outdoor games requiring group engagement [41].

Various measures can be taken by medical institutions and governing bodies like Nepal Medical Association (NMA) and Nepal Medical Council (NMC). They should create a nurturing, learning environment, teach skills for stress management and promote self-awareness. Adjustment in the curriculum and teaching methods can be made to reduce distress among medical students [42]. A trustworthy and stigma-free environment should be built for medical students and health care professionals. They should also be encouraged to seek professional counseling or therapy when in need. Further studies are required to understand distress among medical students and health care professionals, and evidence-based approaches to dealing with such difficulties. Institutions such as the University Grants Commission (UGC) and Nepal Health Research Council (NHRC) should make provisions to fund such future studies. It is also recommended to adapt and validate screening tools for use in the Nepali population in the future [43].

Some limitations of the study are acknowledged. The study was conducted using an online self-reporting questionnaire, yet we had a good response rate of 95.8% (n = 229). Although the selection and response bias was minimized, the social desirability bias might have affected the results of our study. The level of burnout among the academic years may be affected by the final board examinations that are held at different times for the respective year. The study

period was a month after classes resumed post-COVID-19 lockdown. However, multiple factors of the ongoing pandemic may have varying effects on the level of burnout for different individuals. Finally, the results may not be representative of all the medical colleges of Nepal since the participants were only from NAIHS but it will surely start the much-needed conversation on this topic. Furthermore, the role of factors such as gender, age, academic year, extracurricular activities, hosteller, relationship status, payment/scholarship students, sleep deprivation, etc. on burnout needs further exploration.

## Conclusion

The prevalence of burnout among medical students in our study was very high, and no associations of burnout with age, gender, or year of study were found. Hence, effective strategies for the mental well-being of future physicians need to be made by relevant authorities such as psychological help desk, mental health support groups, and amendments in the academic curriculum.

## Supporting information

**S1 File. Corrected IRC proposal.**
(PDF)

**S2 File. OLBI questionnaire.**
(PDF)

**S3 File. Burnout.**
(DTA)

## Author Contributions

**Conceptualization:** Dhan Bahadur Shrestha, Nagendra Katuwal, Ayush Tamang, Agrima Paudel, Anu Gautam, Muna Sharma, Ujwal Bhusal.

**Formal analysis:** Dhan Bahadur Shrestha.

**Methodology:** Ayush Tamang, Agrima Paudel, Anu Gautam, Muna Sharma, Ujwal Bhusal, Pravash Budhathoki.

**Project administration:** Ayush Tamang, Agrima Paudel, Anu Gautam, Muna Sharma, Ujwal Bhusal.

**Supervision:** Dhan Bahadur Shrestha, Nagendra Katuwal.

**Validation:** Dhan Bahadur Shrestha, Nagendra Katuwal.

**Writing – original draft:** Dhan Bahadur Shrestha, Ayush Tamang, Agrima Paudel, Anu Gautam, Muna Sharma, Ujwal Bhusal, Pravash Budhathoki.

**Writing – review & editing:** Dhan Bahadur Shrestha, Nagendra Katuwal, Ayush Tamang, Agrima Paudel, Anu Gautam, Muna Sharma, Ujwal Bhusal, Pravash Budhathoki.

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
