## [Decision Letter · Decision Letter 0]

5 May 2021

PONE-D-21-09730

Burnout among medical students of a medical college in Kathmandu; A cross sectional study

PLOS ONE

Dear Dr. Shrestha,

Thank you for submitting your manuscript to PLOS ONE. After careful consideration, we feel that it has merit but does not fully meet PLOS ONE’s publication criteria as it currently stands. Therefore, we invite you to submit a revised version of the manuscript that addresses the points raised during the review process.

We look forward to receiving your revised manuscript.

Kind regards,

Prof. Ritesh G. Menezes, M.B.B.S., M.D., Diplomate N.B.

Academic Editor

PLOS ONE

Journal Requirements:

'Ethical clearance was obtained from the Institutional Review Committee (IRC) of NAIHS before collecting data from participants (Ref no: 365). All the respondents were informed about the aim and objectives of the study by including the consent form in the questionnaire itself. The participants were aware that their participation was voluntary. Confidentiality of the participants was ensured. '

b) Please provide additional details regarding participant consent. In the ethics statement in the Methods and online submission information, please ensure that you have specified what type you obtained (for instance, written or verbal, and if verbal, how it was documented and witnessed). If your study included minors, state whether you obtained consent from parents or guardians. If the need for consent was waived by the ethics committee, please include this information.

Reviewers' comments:

Reviewer's Responses to Questions

**Comments to the Author**

1. Is the manuscript technically sound, and do the data support the conclusions?

Reviewer #1: Yes

Reviewer #2: Yes

Reviewer #3: Yes

Reviewer #4: Partly

Reviewer #5: Yes

Reviewer #6: Yes

Reviewer #7: Yes

Reviewer #8: Yes

2. Has the statistical analysis been performed appropriately and rigorously? 

Reviewer #1: Yes

Reviewer #2: Yes

Reviewer #3: Yes

Reviewer #4: Yes

Reviewer #5: Yes

Reviewer #6: Yes

Reviewer #7: Yes

Reviewer #8: I Don't Know

3. Have the authors made all data underlying the findings in their manuscript fully available?

Reviewer #1: Yes

Reviewer #2: No

Reviewer #3: Yes

Reviewer #4: No

Reviewer #5: Yes

Reviewer #6: Yes

Reviewer #7: Yes

Reviewer #8: Yes

4. Is the manuscript presented in an intelligible fashion and written in standard English?

Reviewer #1: No

Reviewer #2: Yes

Reviewer #3: Yes

Reviewer #4: Yes

Reviewer #5: Yes

Reviewer #6: Yes

Reviewer #7: Yes

Reviewer #8: Yes

5. Review Comments to the Author

Reviewer #1: I find the manuscript well written and addresses the burnout in medical undergraduates in Nepal. I have the following comments to make:

1. Methods: Please further elaborate on how the authors got the email addresses of the students. Did the authors collect any identifying information from the students (for e.g., email addresses)? How long was the link to the questionnaire made active? Why did it take so long for the students to respond to the questionnaire? Did the authors contact students to fill up the survey form as they didn't receive the adequate response on time? How did the authors know that no student filled out the survey form more than once? As there are co-authors from third year medical school where the study was conducted, it would be wise to mention that none of the co-authors filled up the questionnaire.

2. Methods: line 93- There is duplication of text. Please remove the sentence at line 74-76.

3. Results: line 139- There is duplication of text. Please delete the text at line 139 and start the sentence with the text beginning at 140.

4. There are some errors in English language, some of which are highlighted below:

a. “cross sectional” is generally written as “cross-sectional”.

b. Line 24 and 192: the prevalence

c. Line 27: “The burnout” be written as “Burnout”

d. Line 27: gender, and year of study

e. Line 66: “physicians is” be replaced as “physicians are”

f. Line 73: the Nepalese

g. Line 77: consent form attached to the form itself.

h. Line 91: 228 and

i. Line 101: depersonalization, emotional exhaustion, and

j. Line 106: patient-related

k. Line 119: use past tense. Replace ‘is’ with ‘was’.

l. Line 119-120: scores of

m. Line 134-135: check spacing

n. Line 140: replace ‘completely filled’ with ‘completed’

o. Line 146-149: use the article ‘the’. For e.g., the disengagement, the total score, the exhaustion

p. Line 151: previous mentioned be written as ‘previously mentioned’

q. Line 152-153: Use article the after under for e.g., ‘under the disengaged group’

r. Line 157-158: for the occurrence

s. Line 169: due to the use of

t. Line 171: due to a multitude of factors

u. Line 173: sentence can be reframed as “Our study found the prevalence of burnout to be considerably higher than previous study conducted in Nepal which…”

v. Line 176: due to the timing

w. Line 177: Still, multiple factors

x. Line 180: significant association of gender with burnout in…

y. Line 199: students were

z. Line 205: exposure to it since

aa. Line 192-199: consider using a hyphen in first year, fourth year etc.

bb. Line 237: performed the

Reviewer #2: Review of

Burnout among medical students of a medical college in Kathmandu: A cross sectional study

1. Reviewer’s Overall Comment:

Research on burnout and emotional exhaustion in Nepal for medical professionals and students is increasing. Nevertheless, there is a gap in the study of burnout among preclinical students. The relevancy of the research is well articulated and explained in the Nepali context. This research, scientifically designed and meticulously carried out after following exact research methods, will support the education and medical institutions to look at the burden of medical education on their students. With minor revisions to be addressed, the reviewer recommends publishing this article.

2. Few comments for adjustment:

- Line 31 – please be specific that the researchers had used the English version of the Oldenburg Burnout Inventory adapted for students. It seems that the tool was not translated and culturally adapted in the research process. Replicate the same in line 105.

- In abstract's result section, line 33, please include the total number of respondents (N = 229).

- Line 78, please describe the sampling process and cite the formula you have used to define the sample size for this research. It would be nice to explain, the reasons for choosing the sample size in that way before straightforwardly putting the formula.

- The data were collected using a google form, was there any record of individual data such as the email of the respondents? Did the participants receive any debriefing note with an explanation of the burnout including exhaustion and disengagement? Did the participant receive any guidance to seek support if they wish?

- Line 136 – how do you explain that the confidentiality and anonymity of the participants were ensured? Kindly explain the process from data collection to data analysis and repository phase of the data.

- Line 173, please mention (N = 651) after % of prevalence and response rate of their study.

- After line 197, it would be worth mentioning that the peer and family pressure to be a medical doctor in Nepali society creates pressure on the students.

• Pokhrel, Khadayat, and Tulachan (2020) outlined other possible causes of higher prevalence of burnout in Medical students – ‘the field of medicine not being one’s first priority, following friends to study medicine, parental wish to study medicine, wrongly choosing the specialty, and being demotivated after learning the reality of medicine. The authors, unfortunately, came across a few circumstances in which some medical students became demotivated after living the reality of medicine. Most of them followed their parents’ wish or dreamt of their higher social status before enrolling into the medical school (p.15).’

The reviewer recommends highlighting such findings in the discussion section.

- Authors have suggested some of the strategies and actions to address the burnout of medical students in the conclusion section. Line 223-225 says, ‘Hence, effective strategies for the mental well-being of future physicians need to be made by relevant authorities such as psychological help desk, mental health support groups, and amendments in the academic curriculum’. It is understandable that this study limits after knowing the prevalence of burnout, nevertheless, it would be great to review secondary literature on effective and efficacious actions for the prevention of burnout in medical students and describe it in the discussion section for future researchers and academia leaders. Here are some of the resources available to similar populations to address burnout of medical students:

• Adhikari (2018) on Compassion fatigue into the Nepali counselors: challenges and recommendations

• Recommendation section of Adhikari (2020) pages – 131 to 134,

• Equally, the study of Adhikari et al. (2017 p.6) mentions, ‘Various solutions to reduce stress like creating a nurturing learning environment, identifying and assisting students, teaching skills for stress management and promoting self-awareness have been proposed. Integration of such programs as a part of medical education with special focus on preclinical students and female students may improve mental health among students’.

 

References

Pokhrel, N. B., Khadayat, R., & Tulachan, P. (2020). Depression, anxiety, and burnout among medical students and residents of a medical school in Nepal: a cross-sectional study. BMC psychiatry, 20(1), 1-18. https://doi.org/10.1186/s12888-020-02645-6

Adhikari, A., Dutta, A., Sapkota, S., Chapagain, A., Aryal, A., & Pradhan, A. (2017). Prevalence of poor mental health among medical students in Nepal: a cross-sectional study. BMC medical education, 17(1), 1-7. https://doi.org/10.1186/s12909-017-1083-0

Adhikari, Y. (2020). Prevalence of Professional Quality of Life (ProQOL) and Its Influence on the Personal Distress of Doctors in Nepal. (Doctor of Philosophy Dissertation, University of Nicosia). https://www.researchgate.net/publication/344397792_Prevalence_of_Professional_Quality_of_Life_ProQOL_and_Its_Influence_on_the_Personal_Distress_of_Doctors_in_Nepal

Adhikari, Y. (2018). Compassion fatigue into the Nepali counselors: Challenges and recommendations. MOJ Public Health, 7(6), 376–379. http://dx.doi.org/10.15406/mojph.2018.07.00271

Reviewer #3: Manuscript is written in important area. Overall flow of content also seems good. Some of my observations are as follows:

Manuscript lacks the clarity while talking about sampling procedure. It is mentioned that there was stratified sampling technique, which is one of the random sampling technique. How could authors made random sampling through google form is not clearly mentioned.

You could add "Model adequacy test (hosmer and lemshw, cox-snell r2" as there is Binary logistic regression.

It would be more appropriate if you make age categories and do multivariate analysis.

No need to keep that many ZERO in Odds ratio.

If your tool supports, it is better to keep your outcome variable into Mild, Moderate and Severe burnout.

Reviewer #4: Authors are more experience of research, Article is current more relevant and Ethical approval letter and Participants consent was obtained. provide a specific recommendation from student side and national medical council level.

Reviewer #5: This is a well written paper that has examined the questions set out in the investigation as well as preformed a rigorous statistical review. The discussion also examines the issues at hand and explains the various factors associated with burnout.

Reviewer #6: This was a very interesting piece to read. The authors have addressed appropriately to the main objective of the study and have also listed all the possible limitations that could have been the cause of the noted higher burnout rates compared to other similar studies.

The authors have referenced a recent similar study published from Nepal, as that was just recently published and had looked at many more variables, I would like the authors to look for any additional data that they might have collected to make this particular study unique.

1) It would be interesting to have a general demographic of the students of each year of study.

2) Line 183 - 'similar/contrary' it would be better to split them in two different sentences.

3) Just curious if the 3 yr, 4th yr and 5th yr students were helping out at a COVID ward, as they would have started clinical rotations and might have been roped in to help out - if do include that too.

Best wishes with your submission.

Reviewer #7: Dear author,

Congratulation for the excellent work and bringing this issue forward. The paper has been written in a nice way with good explanation of methodology and in depth analysis where possible. Few queries

1. You have calculated Cronbach's Alpha for the tool used, which is already a validated tool. Did you have any modification or translation of this tool.

2. Selection of Target population, N = 560* in sample size calcualtion?? Explain

3. More literature review and more elaborated discussion section from the regional or global literature is expected if there is less literature from Nepal.

4. Explain the reason for choosing only selected variable (years, gender and year of study) for binary logistic regression and its significance.

5. Biases in self administered questionnaires during the use of Likert Scale? Please see

Reviewer #8: Summary of the research

This was the cross-sectional study about burnout of medical student in a medical college in Kathmandu, Nepal. Methodology of the study appears sound and study designs is appropriate for the study. Instrument used was Oldenburg Burnout Inventory adapted for students (OLBI-S) which has been shown to have good internal consistency. As per my (poor) knowledge on statistics, I don’t find any fault in statistical analysis. Results shows high prevalence of burnout among medical students which is mentioned in the conclusion. All data are accessible. References are ok. Title is appropriate and informative.

Major issues observed: None.

Minor issues:

Abstract: Authors should clarify whether it is burden or burnout (line 24) and put a comma or restructure the sentence to make it understandable (line 33/34).

Introduction: Please cross check the spellings (line 68)

Study sample: Authors need to make it clear how many class years were taken as stratum (line 95)

Study instruments: Authors need to focus on the instrument they are using with mention of pros and cons of using it rather than describing about instruments they are not using (line 100-107).

Analytical strategy: internal consistency of the instrument used can be put in the methodology parts rather than in analysis part. (line 129-131)

Conclusion: Can we put it as ‘The prevalence of burnout among medical students in our study was very high’ or something like this? If we write ‘More than expected’, authors may need to clarify what was expected (line 221).

I suggest the authors should hire a professional English language/academic editors to get help in editing the write up.

Review Conclusion:

The study topic is an interesting one. The study is methodologically sound. Conclusions are well presented. However, it needs minor revision in areas of grammars and editing part.

6. PLOS authors have the option to publish the peer review history of their article (what does this mean?). If published, this will include your full peer review and any attached files.

Reviewer #1: **Yes: **Alok Atreya

Reviewer #2: **Yes: **Yubaraj Adhikari

Reviewer #3: **Yes: **Dr. Kishor Adhikari

Reviewer #4: **Yes: **Rajesh Kumar Yadav

Reviewer #5: **Yes: **Dr. Rijen Shrestha

Reviewer #6: No

Reviewer #7: No

Reviewer #8: No

---

## [Author Response · Author response to Decision Letter 0]

7 May 2021

Joerg Heber, PhD

Editor in Chief,

PLOS ONE

San Francisco, California, US

Dear Dr. Heber,

Referring to our study “PONE-D-21-09730: Burnout among medical students of a medical college in Kathmandu; A cross sectional study”, thank you for providing such an opportunity to revise and edit our manuscript to make it more clear and understandable to audience. We author believe, after these revision, now our manuscript is more clear and replicable to enthusiast and researcher of such title. And, we believe this will help to modify curricula and help policy makers to review policy/curricula for medical students and health personals to decrease burnout and ease their day to day life. 

We have revised our manuscript as per the comments by reviewers and here we have responded to every comments by editor and reviewer in point to point fashion. We have mentioned reply to every comments as appropriate. 

Reviews from academic editor: 

https://journals.plos.org/plosone/s/file?id=ba62/PLOSOne_formatting_sample_tite_authors_affiliations.pdf

Reply: Thank you for the comment. We have revised and adhered to PLOS guidelines

'Ethical clearance was obtained from the Institutional Review Committee (IRC) of NAIHS before collecting data from participants (Ref no: 365). All the respondents were informed about the aim and objectives of the study by including the consent form in the questionnaire itself. The participants were aware that their participation was voluntary. Confidentiality of the participants was ensured. '

Reply: Thank you for the comment. We have amended our ethics statement to include the full name of the ethics committee that approved our study.

b) Please provide additional details regarding participant consent. In the ethics statement in the Methods and online submission information, please ensure that you have specified what type you obtained (for instance, written or verbal, and if verbal, how it was documented and witnessed). If your study included minors, state whether you obtained consent from parents or guardians. If the need for consent was waived by the ethics committee, please include this information.

Reply: Thank you for the comment. We have specified the type of consent from the participants.

Reply: Thank you for the comment. Amended

Reviewer 1: 

1. I find the manuscript well written and addresses the burnout in medical undergraduates in Nepal. I have the following comments to make:

1. Methods: Please further elaborate on how the authors got the email addresses of the students. Did the authors collect any identifying information from the students (for e.g., email addresses)? How long was the link to the questionnaire made active? Why did it take so long for the students to respond to the questionnaire? Did the authors contact students to fill up the survey form as they didn't receive the adequate response on time? How did the authors know that no student filled out the survey form more than once? As there are co-authors from third year medical school where the study was conducted, it would be wise to mention that none of the co-authors filled up the questionnaire.

Reply: Thank you for the comment. Methods: We have made necessary changes mentioned by reviewer 1 which includes how we got the email address of the participants, if we take any identifying information from participants. Similarly, we have explained how it took a long period of time for participants to respond and how we have made sure that single participants provide a single response. Yes, there are co-author from third year and they were not included in the study.

2. Methods: line 93- There is duplication of text. Please remove the sentence at line 74-76.

Reply: Thank you for the comment. We corrected the duplication of text at line 74-75.

3. Results: line 139- There is duplication of text. Please delete the text at line 139 and start the sentence with the text beginning at 140.

Reply: Thank you for the comment. Results: We corrected the duplication of text as mentioned.

4. 4. There are some errors in English language, some of which are highlighted below:

a. “cross sectional” is generally written as “cross-sectional”.

b. Line 24 and 192: the prevalence

c. Line 27: “The burnout” be written as “Burnout”

d. Line 27: gender, and year of study

e. Line 66: “physicians is” be replaced as “physicians are”

f. Line 73: the Nepalese

g. Line 77: consent form attached to the form itself.

h. Line 91: 228 and

i. Line 101: depersonalization, emotional exhaustion, and

j. Line 106: patient-related

k. Line 119: use past tense. Replace ‘is’ with ‘was’.

l. Line 119-120: scores of

m. Line 134-135: check spacing

n. Line 140: replace ‘completely filled’ with ‘completed’

o. Line 146-149: use the article ‘the’. For e.g., the disengagement, the total score, the exhaustion

p. Line 151: previous mentioned be written as ‘previously mentioned’

q. Line 152-153: Use article the after under for e.g., ‘under the disengaged group’

r. Line 157-158: for the occurrence

s. Line 169: due to the use of

t. Line 171: due to a multitude of factors

u. Line 173: sentence can be reframed as “Our study found the prevalence of burnout to be considerably higher than previous study conducted in Nepal which…”

v. Line 176: due to the timing

w. Line 177: Still, multiple factors

x. Line 180: significant association of gender with burnout in…

y. Line 199: students were

z. Line 205: exposure to it since

aa. Line 192-199: consider using a hyphen in first year, fourth year etc.

bb. Line 237: performed the

Reply: Thank you for such a detailed corrections advised. Errors in English language mentioned were also dealt accordingly.

Reviewer 2:

1. Reviewer’s Overall Comment:

Research on burnout and emotional exhaustion in Nepal for medical professionals and students is increasing. Nevertheless, there is a gap in the study of burnout among preclinical students. The relevancy of the research is well articulated and explained in the Nepali context. This research, scientifically designed and meticulously carried out after following exact research methods, will support the education and medical institutions to look at the burden of medical education on their students. With minor revisions to be addressed, the reviewer recommends publishing this article.

Reply: Thank you for your comment/feedback. 

2. Few comments for adjustment:

- Line 31 – please be specific that the researchers had used the English version of the Oldenburg Burnout Inventory adapted for students. It seems that the tool was not translated and culturally adapted in the research process. Replicate the same in line 105. 

Reply: Line 31: We have specified that we used the English version of Oldenburg Burnout Inventory (OLBI). 

- In abstract's result section, line 33, please include the total number of respondents (N = 229).

Reply: Line 33: We have included the total number of respondents.

- Line 78, please describe the sampling process and cite the formula you have used to define the sample size for this research. It would be nice to explain, the reasons for choosing the sample size in that way before straightforwardly putting the formula.

Reply: Line 78: We have described the sampling process and cited the formula we have used to define the sample size.

- The data were collected using a google form, was there any record of individual data such as the email of the respondents? Did the participants receive any debriefing note with an explanation of the burnout including exhaustion and disengagement? Did the participant receive any guidance to seek support if they wish?

Reply: Yes, there was a record of individual data by a unique random number assigned to each participant. We hadn't made any provision of debriefing notes with explanation of the burnout and its scales Disengagement and Exhaustion. We didn't inform participants about their burnout scores. 

- Line 136 – how do you explain that the confidentiality and anonymity of the participants were ensured? Kindly explain the process from data collection to data analysis and repository phase of the data.

Reply: Line 136- The responses from Google Forms didn't record individual names or their email addresses to maintain anonymity of the participants.

- Line 173, please mention (N = 651) after % of prevalence and response rate of their study.

Reply: Line 173- we have mentioned the total population size(N = 560)

- After line 197, it would be worth mentioning that the peer and family pressure to be a medical doctor in Nepali society creates pressure on the students.

 Pokhrel, Khadayat, and Tulachan (2020) outlined other possible causes of higher prevalence of burnout in Medical students – ‘the field of medicine not being one’s first priority, following friends to study medicine, parental wish to study medicine, wrongly choosing the specialty, and being demotivated after learning the reality of medicine. The authors, unfortunately, came across a few circumstances in which some medical students became demotivated after living the reality of medicine. Most of them followed their parents’ wish or dreamt of their higher social status before enrolling into the medical school (p.15).’

The reviewer recommends highlighting such findings in the discussion section.

Reply: As pointed out by the reviewer, we had forgotten the "peer and family pressure" to be a medical doctor in Nepali society. We have included that part in our discussion. "Some students might have joined medical school under parental or peer pressure, or with the desire of higher socioeconomic status. They may become demotivated after knowing the reality of medicine and develop more negative attitudes towards medicine as a career."

Authors have suggested some of the strategies and actions to address the burnout of medical students in the conclusion section. Line 223-225 says, ‘Hence, effective strategies for the mental well-being of future physicians need to be made by relevant authorities such as psychological help desk, mental health support groups, and amendments in the academic curriculum’. It is understandable that this study limits after knowing the prevalence of burnout, nevertheless, it would be great to review secondary literature on effective and efficacious actions for the prevention of burnout in medical students and describe it in the discussion section for future researchers and academia leaders. Here are some of the resources available to similar populations to address burnout of medical students:

• Adhikari (2018) on Compassion fatigue into the Nepali counselors: challenges and recommendations

• Recommendation section of Adhikari (2020) pages – 131 to 134,

• Equally, the study of Adhikari et al. (2017 p.6) mentions, ‘Various solutions to reduce stress like creating a nurturing learning environment, identifying and assisting students, teaching skills for stress management and promoting self-awareness have been proposed. Integration of such programs as a part of medical education with special focus on preclinical students and female students may improve mental health among students’.

Reply: Regarding strategies and actions to address burnout among medical students, we went through the sources mentioned by you and have made changes in our discussion section. We have included recommendations for students, institutions and governing bodies. 

Reviewer 3:

Manuscript is written in important area. Overall flow of content also seems good. Some of my observations are as follows:

Manuscript lacks the clarity while talking about sampling procedure. It is mentioned that there was stratified sampling technique, which is one of the random sampling technique. How could authors made random sampling through google form is not clearly mentioned.

Reply: We had generated the random numbers using random number generator software online, and then the respective random numbers were sent to the participants via mail along with Google forms. 

You could add "Model adequacy test (hosmer and lemshw, cox-snell r2" as there is Binary logistic regression.

Reply: Our model cox-snell r2 was 0.0216

It would be more appropriate if you make age categories and do multivariate analysis.

Reply: Being age range was small we prefer not to categorise further due to small range among five batches of medical students,

No need to keep that many ZERO in Odds ratio.

Reply: Amended

If your tool supports, it is better to keep your outcome variable into Mild, Moderate and Severe burnout.

Reply: Pardon us, but the tool we used has interpretation as in Figure 1 only, so we could not categorize into into Mild, Moderate and Severe burnout.

Reviewer 4: 

Authors are more experience of research, Article is current more relevant and Ethical approval letter and Participants consent was obtained. provide a specific recommendation from student side and national medical council level.

Reply: As pointed out by the reviewer, we have added specific recommendations from both the student side as well as national medical council level.

Reviewer 5

This is a well written paper that has examined the questions set out in the investigation as well as preformed a rigorous statistical review. The discussion also examines the issues at hand and explains the various factors associated with burnout.

Reply: Thank you for the comment

Reviewer 6

This was a very interesting piece to read. The authors have addressed appropriately to the main objective of the study and have also listed all the possible limitations that could have been the cause of the noted higher burnout rates compared to other similar studies.

The authors have referenced a recent similar study published from Nepal, as that was just recently published and had looked at many more variables, I would like the authors to look for any additional data that they might have collected to make this particular study unique.

1) It would be interesting to have a general demographic of the students of each year of study.

Reply: 

2) Line 183 - 'similar/contrary' it would be better to split them in two different sentences.

Reply: Line 183: We have splitted the sentence into two

3) Just curious if the 3 yr, 4th yr and 5th yr students were helping out at a COVID ward, as they would have started clinical rotations and might have been roped in to help out - if do include that too.

Reply: Thank you for your comment. No, there were no any students from 3rd, 4th or 5th year helping out at a COVID ward.

Reviewer 7

Congratulation for the excellent work and bringing this issue forward. The paper has been written in a nice way with good explanation of methodology and in depth analysis where possible. Few queries

You have calculated Cronbach's Alpha for the tool used, which is already a validated tool. Did you have any modification or translation of this tool.

Reply: No, we had not done any modification or translation of this tool. 

Selection of Target population, N = 560* in sample size calcualtion?? Explain

Our target population is the total number of medical students from first to fifth year in our college. Hence, N= 560. 

More literature review and more elaborated discussion section from the regional or global literature is expected if there is less literature from Nepal.

Reply: We have tried to include the literature review from the regional and global literature. Here are link to some of our references Nepal, Saudi Arabia, Lebanon, USA,, Brazil, Cyprus

Explain the reason for choosing only selected variables (years, gender and year of study) for binary logistic regression and its significance.

Reply: Those were the distinct variables among medical students which could have affected the dependent variables so taken as independent variables for binary regression. 

Biases in self-administered questionnaires during the use of Likert Scale? Please see

Reply: Kept as limitation 

Reviewer 8

Summary of the research

This was the cross-sectional study about burnout of medical student in a medical college in Kathmandu, Nepal. Methodology of the study appears sound and study designs is appropriate for the study. Instrument used was Oldenburg Burnout Inventory adapted for students (OLBI-S) which has been shown to have good internal consistency. As per my (poor) knowledge on statistics, I don’t find any fault in statistical analysis. Results shows high prevalence of burnout among medical students which is mentioned in the conclusion. All data are accessible. References are ok. Title is appropriate and informative.

Major issues observed: None.

Minor issues:

Abstract: Authors should clarify whether it is burden or burnout (line 24) and put a comma or restructure the sentence to make it understandable (line 33/34).

Reply: Abstract: Line 24: The "burden" was corrected to "burnout"

Line 33/34: The sentence was restructured.

Introduction: Please cross check the spellings (line 68)

Reply: Introduction: Line 68: We cross checked the spellings

Study sample: Authors need to make it clear how many class years were taken as stratum (line 95)

Reply: Study sample: Line 95: 5 class years from 1st year to 2nd year were taken, and we have mentioned that in the manuscript

Study instruments: Authors need to focus on the instrument they are using with mention of pros and cons of using it rather than describing about instruments they are not using (line 100-107).

Reply: Study instruments: Line 100-107: We had mentioned some pros and cons of the instrument used(OLBI), to which we have now added some.

Analytical strategy: internal consistency of the instrument used can be put in the methodology parts rather than in analysis part. (line 129-131)

Reply: Analytical strategy: Line 129-13: Internal consistency of the instrument used was calculated by ourselves, so it was included in the analysis part rather than the methodology, as it would have referred to the internal consistency calculated by the authors of this instrument 

Conclusion: Can we put it as ‘The prevalence of burnout among medical students in our study was very high’ or something like this? If we write ‘More than expected’, authors may need to clarify what was expected (line 221).

Reply: Conclusion: line 221: We restructured the sentence as you suggested

Review Conclusion:

The study topic is an interesting one. The study is methodologically sound. Conclusions are well presented. However, it needs minor revision in areas of grammars and editing part.

Reply: Thank you for your comments and suggestions. We have amended and edited language and grammar where required.

---

## [Decision Letter · Decision Letter 1]

7 Jun 2021

PONE-D-21-09730R1

Burnout among medical students of a medical college in Kathmandu; A cross-sectional study

PLOS ONE

Dear Dr. Shrestha,

Thank you for submitting your manuscript to PLOS ONE. After careful consideration, we feel that it has merit but does not fully meet PLOS ONE’s publication criteria as it currently stands. Therefore, we invite you to submit a revised version of the manuscript that addresses the points raised during the review process.

We look forward to receiving your revised manuscript.

Kind regards,

Prof. Ritesh G. Menezes, M.B.B.S., M.D., Diplomate N.B.

Academic Editor

PLOS ONE

Journal Requirements:

Additional Academic Editor Comments:

- Comments by Reviewer #4 have not been completely addressed in the revised manuscript. Kindly check the attachments for detailed comments made by Reviewer #4 and provide a specific point-to-point response.

Reviewers' comments:

Reviewer's Responses to Questions

**Comments to the Author**

1. If the authors have adequately addressed your comments raised in a previous round of review and you feel that this manuscript is now acceptable for publication, you may indicate that here to bypass the “Comments to the Author” section, enter your conflict of interest statement in the “Confidential to Editor” section, and submit your "Accept" recommendation.

Reviewer #1: All comments have been addressed

Reviewer #2: All comments have been addressed

Reviewer #3: All comments have been addressed

Reviewer #4: All comments have been addressed

Reviewer #6: All comments have been addressed

Reviewer #8: All comments have been addressed

2. Is the manuscript technically sound, and do the data support the conclusions?

Reviewer #1: (No Response)

Reviewer #2: Yes

Reviewer #3: Yes

Reviewer #4: Yes

Reviewer #6: Yes

Reviewer #8: Yes

3. Has the statistical analysis been performed appropriately and rigorously? 

Reviewer #1: (No Response)

Reviewer #2: Yes

Reviewer #3: Yes

Reviewer #4: Yes

Reviewer #6: Yes

Reviewer #8: Yes

4. Have the authors made all data underlying the findings in their manuscript fully available?

Reviewer #1: (No Response)

Reviewer #2: Yes

Reviewer #3: Yes

Reviewer #4: Yes

Reviewer #6: Yes

Reviewer #8: Yes

5. Is the manuscript presented in an intelligible fashion and written in standard English?

Reviewer #1: (No Response)

Reviewer #2: Yes

Reviewer #3: Yes

Reviewer #4: Yes

Reviewer #6: Yes

Reviewer #8: Yes

6. Review Comments to the Author

Reviewer #1: (No Response)

Reviewer #2: Authors have addressed all recomendations and revised the paper accordingly. Thanks to authors for incorporation of my comments.

The reviewer has no further comments on the revised paper.

I believe, this paper helps to sensitize the burnout on future doctors and act accordingly to reduce the burden.

Reviewer #3: (No Response)

Reviewer #4: Dear Author,

Everything seems fine. You have addressed all those comments. Best wishes for your publications!!!

Thank you !!!

Reviewer #6: Good work on answering most of the comments and great job on the quick turn around. I have no additional comments.

Reviewer #8: (No Response)

7. PLOS authors have the option to publish the peer review history of their article (what does this mean?). If published, this will include your full peer review and any attached files.

Reviewer #1: **Yes: **Alok Atreya

Reviewer #2: **Yes: **Yubaraj Adhikari, PhD

Reviewer #3: **Yes: **Dr. Kishor Adhikari

Reviewer #4: **Yes: **Rajesh Kumar Yadav

Reviewer #6: No

Reviewer #8: **Yes: **Dr Khagendra Kafle

---

## [Author Response · Author response to Decision Letter 1]

7 Jun 2021

Joerg Heber, PhD

Editor in Chief,

PLOS ONE

San Francisco, California, US

Dear Dr. Heber,

Referring to our study “PONE-D-21-09730: Burnout among medical students of a medical college in Kathmandu; A cross-sectional study”, thank you for providing such an opportunity to revise and edit our manuscript to make it more clear and understandable to the audience. We author believe, after these revisions, now our manuscript is more clear and replicable to enthusiasts and researchers of such title. And, we believe this will help to modify curricula and help policymakers to review policy/curricula for medical students and health personals to decrease burnout and ease their day-to-day life. 

We have revised our manuscript as per the comments by reviewers and here we have responded to every comment by editor and reviewer in a point-to-point fashion. We have mentioned replies to every comment as appropriate. 

Journal Requirements:

Reply: For easy handling of references we use the automatic Mendeley plugin of PLOS ONE. So, we assume the referencing style is as per journal requirements.

Additional Academic Editor Comments:

- Comments by Reviewer #4 have not been completely addressed in the revised manuscript. Kindly check the attachments for detailed comments made by Reviewer #4 and provide a specific point-to-point response.

Reply: Thank you for noting the comment. We have already amended that comment in the manuscript. But pardon us for not keeping whole changes in response. We did not find any new comments to amend further. Please find a specific response as it is done in the manuscript in the reply section of the specific comment.

Reviewer 4: 

Authors are more experience of research, Article is current more relevant and Ethical approval letter and Participants consent was obtained. provide a specific recommendation from student side and national medical council level.

Reply: As pointed out by the reviewer, we have added specific recommendations from both the student side as well as national medical council level. It has been added and described already in previously revised submission in later part of discussion section as “At an individual level, students should be encouraged to incorporate various self-care techniques into their daily lives such as a nutritious diet, regular exercise, a restful sleep, making a balance between study and leisure, the practice of self-compassion, and being aware of their emotional needs [40]. Similarly, positive coping and hobbies help to overcome the stressors of medical studies. Some common practices can be mindfulness, yoga, listening to music, reading books, and outdoor games requiring group engagement [41]. 

Various measures can be taken by medical institutions and governing bodies like Nepal Medical Association (NMA) and Nepal Medical Council (NMC). They should create a nurturing, learning environment, teach skills for stress management and promote self-awareness. Adjustment in the curriculum and teaching methods can be made to reduce distress among medical students [42]. A trustworthy and stigma-free environment should be built for medical students and health care professionals and encouraged to seek professional counseling or therapy when in need. Further studies are required to understand distress among medical students and health care professionals, and evidence-based approaches to dealing with such difficulties. Institutions such as the University Grants Commission (UGC) and Nepal Health Research Council (NHRC) should make provisions to fund such future studies. It is also recommended to adapt and validate screening tools for use in the Nepali population in the future [43].”

We did not notice any new comments for more action. If any corrections are warranted, will be happy to amend them. 

Thank you

---

## [Editor Report · Decision Letter 2]

14 Jun 2021

Burnout among medical students of a medical college in Kathmandu; A cross-sectional study

PONE-D-21-09730R2

Dear Dr. Shrestha,

We’re pleased to inform you that your manuscript has been judged scientifically suitable for publication and will be formally accepted for publication once it meets all outstanding technical requirements.

Kind regards,

Prof. Ritesh G. Menezes, M.B.B.S., M.D., Diplomate N.B.

Academic Editor

PLOS ONE

---

## [Editor Report · Acceptance letter]

16 Jun 2021

PONE-D-21-09730R2 

Burnout among medical students of a medical college in Kathmandu; A cross-sectional study 

Dear Dr. Shrestha:

I'm pleased to inform you that your manuscript has been deemed suitable for publication in PLOS ONE. Congratulations! Your manuscript is now with our production department. 

Kind regards, 

on behalf of

Prof. Dr. Ritesh G. Menezes 

Academic Editor

PLOS ONE